# Multi-Instance Multi-Label Learning for Text-motion Retrieval

## ABSTRACT

Text-motion retrieval (TMR) is a significant cross-modal task that retrieves motion sequences semantically similar to a given query text. Existing TMR methods primarily utilize single embeddings to represent and align text and motion sequences. However, real-world motion sequences typically contain multiple atomic motions with complex semantics, which is hard to precisely capture by single embeddings. Additionally, the common co-occurring and coupling of atomic motions further post significant challenges in effective modeling and aligning text and motion sequences. In this paper, we regard TMR as a Multi-Instance Multi-Label (MIML) learning problem, where the motion sequence is viewed as a bag of atomic motions and the text is the bag of corresponding phrases. To address the MIML problem, we propose a novel Multi-Granularity Semantics Interaction (MGSI) approach, which effectively captures and aligns the semantics of text and motion sequences across various levels. Specifically, the MGSI approach initially decomposes both the query and motion sequences into three hierarchical levels: token, instance, and bag. Then, we utilize graph neural networks to explicitly model their semantics correlation and perform semantics interaction at these respective levels, precisely capturing the semantics at multiple granularities. To identify and model co-occurring atomic motions, we measure the frame-wise semantic consistency between motions and then fuse and interact the accordant ones to refine their representations. Finally, we exploit token, instance, and bag-wise semantics interaction to comprehensively align text and motions sequence. We evaluated our methods on two widely-used benchmark datasets, HumanML3D and KIT-ML. The proposed method achieves significant improvements, outperforming the state-of-the-art with a 23.09% increase in Rsum on HumanML3D and a 21.84% increase on KIT-ML.

## CCS CONCEPTS

• **Information systems → Multimedia and multimodal retrieval**.

## KEYWORDS

Text-motion Retrieval, Multi-modal, Cross-modal Alignment

## 1 INTRODUCTION

With the tremendous growth of motion generation tools and methods [7, 17, 22, 26, 27], millions of motion data advent to the world. The ability to efficiently retrieve specific motion sequences from

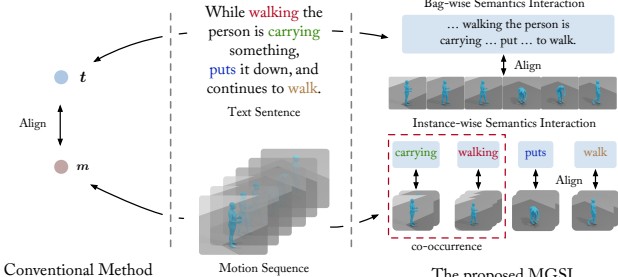

**Figure 1: The comparison between the conventional method and the proposed Multi-Granularity Semantics Interaction (MGSI) framework for TMR. Different from the conventional method that represents text and motion sequence with single point embeddings to alignment, we align the query text and the motion sequences in a hierarchical level.**

this vast repository has become an increasingly critical need. Text-motion retrieval (TMR) [15] is a typical multi-modal retrieval task that aims to retrieve semantically similar motion sequence by the given query text. Following the paradigms of conventional multi-modal retrieval, *e.g.*, text-image and text-video retrieval, numerous TMR researches [4, 12, 14, 15, 21, 24] are raised. Tevet *et al*. [21] introduce a MotionCLIP, where a motion auto-encoder is trained not only to reconstruct motion sequences but also to align their latent representations with the corresponding textual and visual representations in the CLIP space [18]. Mathis *et al*. [15] propose the task of text-motion retrieval and establish a series of evaluation benchmarks with varying difficulty and introduce a joint synthesis and retrieval framework. Yan *et al*. [24] adopt a dual-unimodal transformer encoder to enable a wide range attention in text and motion sequence and introduce a drop triplet loss function to mine the false negative samples.

Although existing work has achieved promising retrieval performance, it generally represents both the query text and motion sequences with a single embedding for alignment. However, in text-motion retrieval, the text and motion sequence typically contain multiple instance, *i.e.*, atomic motions in a motion sequence, verb phrases in a sentence, and include distinct semantics instead of the single samples with unique semantic. As shown in Fig. 1, the text-motion pair contains three atomic motions, *i.e.*, "walking", "carrying" and "put down". The conventional methods may fall short in accurately modeling the complicated semantics of these atomic motions due to the simple representation approach. In addition, these multiple atomic motions are usually co-occurring and overlapping with each other. As shown in Fig. 1, the motion "walking" and "carrying" are co-occurring and overlapping with each other. Solely aligning the single representation of query text and motion sequence may struggle to achieve a accurate cross-modal relation matching in these co-occurring motions, degenerating the retrieval performance. It is necessary to develop a effective method that

explicitly represents and accurately aligns multiple atomic motions and corresponding text.

In this work, we argue that the text-motion retrieval can be viewed as a classical learning problem, Multi-Instance Multi-Label (MIML) [32], where sample is defined as a bag of multiple instances and associated with multiple class label. Thus, we regard the text-motion retrieval task as the MIML learning problem, in which the motion sequence is a bag of atomic motions and the query text is a bag of phrases corresponding to atomic motions. To address the MIML problem of TMR, we propose a Multi-Granularity Semantics Interaction (MGSI) approach as shown in right of Fig. 1. Specifically, in MGSI, we start by decomposing text and motion sequences into three hierarchical levels: token, instance, and bag to represent various granularity semantics. We employ graph neural networks to build a text and motion graph, where the noun phrases, verbs, and sentence (frames, atomic motions, and sequence) are viewed as the token, instance, bag-wise nodes, respectively. Then, we propose a novel co-occurrence motions mining approach that measures the semantics consistency in frame-wise to score the atomic motions. With the consistency score, the co-occurring atomic motions could be identified and fused to generate the co-occurrence features for update the instance nodes. The graph reasoning is applied on the update graph to capture the complex relationships among these components effectively. After that, we introduce a semantic interaction in token, instance, and bag-wise to migrate the semantics correlation between text and motion sequence, achieving a precise cross-modal alignment. Comprehensive evaluations conducted on two widely used benchmark datasets, HumanML3D and KIT-ML, demonstrate that the proposed MGSI surpasses the state-of-the-art methods in a clear margin.

The main contributions of our work are summarised as follows:

- In this work, we formulate the text-motion retrieval as a Multi-Instance Multi-Label (MIML) learning problem, where text sequences are treated as a bag of verbs and motion sequences as a bag of atomic motion instances. To the best of our knowledge, this is the first attempt to model the text-motion retrieval as MIML problem.
- We propose a novel multi-granularity semantics interaction (MGSI) approach to address the MIML problem of TMR, in which we exploit the graph neural networks to decompose the text and motion sequences into token, instance, and bag and perform cross-modal semantics interaction in the corresponding granularity to enable a precise cross-modal alignment.
- Extensive experiments on two widely-used benchmark datasets, HumanML3D [4] and KIT-ML [16], demonstrate that our proposed method surpasses the state-of-the-art, achieving a 23.09% increase in Rsum on HumanML3D and a 21.84% increase on KIT-ML.

## 2 RELATED WORK

### 2.1 Text-motion retrieval

Text-motion retrieval is received much attention in recent years [12, 15], which aims to retrieve semantics relevant motion sequences by a given natural language. Different from conventional cross-modal retrieval [6, 23, 28, 29, 31], the TMR is a challenging task due to the

sequence involving multiple motions with complex semantics. In general, existing TMR methods [4, 12, 14, 15, 21, 24] follow text-to-image or text-to-video retrieval, representing query and motion sequences as single-point embeddings in a common space to be retrieved based on their distance. Guo *et al.* [4] adopts the triplet loss function to perform the motion retrieval, which is used for evaluate the synthesis models. Mathis *et al.* [15] firstly establish the text to 3D human motion retrieval as a standalone task. They simply extend the state-of-the-art text-motion synthesis model TEMOS [14] to TMR by employing the contrastive learning widely used in information retrieval. Nicola *et al.* [12] investigate the content-based large volumes of spatio-temporal skeleton data retrieval by exploiting the transformer-based approach that consists of a ViViT-based motion encoder and CLIP-based [18] text encoder. Yan *et al.* [24] investigate the false negative samples that semantically similar to the anchor but are defined as the negative samples in TMR and propose a drop triplet loss function to calibrate the supervision provided by these false negative samples. However, these methods primarily focus on representing the motion sequence with complex semantics to a global representation for alignment. It inadequately captures the diverse semantics within the motion sequence and hardly enables a comprehensive cross-modal alignment.

### 2.2 Multi-instance multi-label learning

Multi-instance multi-label learning (MIML) [32] is a classical learning problem that is close to the real-world scenarios. Different from the multi-instance learning [2] and multi-label learning [30], the MIML is a more general problem. In MIML, a sample is defined as a bag of multiple instances and associated with multiple class labels. Yang *et. al* [25] introduce the MIML into the privileged information and propose a MIML-FCN+ network to utilize the readily available privileged bags, making the system more general and practical in real world applications. Pan *et. al* [13] view the semi-supervised automatic waveform recognition as a MIML problem and propose a MIML-GAN in which a GAN is incorporated to MIML principle to establish the adversarial learning structure, through which the generator and the discriminator alternatively improve their feature representation and classification abilities, respectively. Lai *et. al* [9] introduce MIML into medical image classification and propose a broad multi-instance multi-label learning to jointly learn multiple sub-networks in a broad sense so that the diverse correlations between bags, instances, and labels can be simultaneously captured. In this work, we regard the text-motion retrieval as MIML problem and propose a multi-granularity semantics interaction that explicitly disentangle the text and motion into token, instance, and bag and exploits the graph neural networks to model correlation of these components. Through applying semantics interaction in corresponding granularity, the MGSI achieves state-of-the-art retrieval performance in two benchmark datasets.

## 3 PRELIMINARY

Given a training dataset $\mathcal{D} = \{(T_i, M_i)\}_{i=1}^{N}$ with $N$ text-motion pairs, text-motion retrieval (TMR) aims to retrieve the semantics relevant motion sequences with the query text (for clarity, we omit the sample index in the following sections). Conventional TMR represents the text and motion as the single point features $t$ and $m$

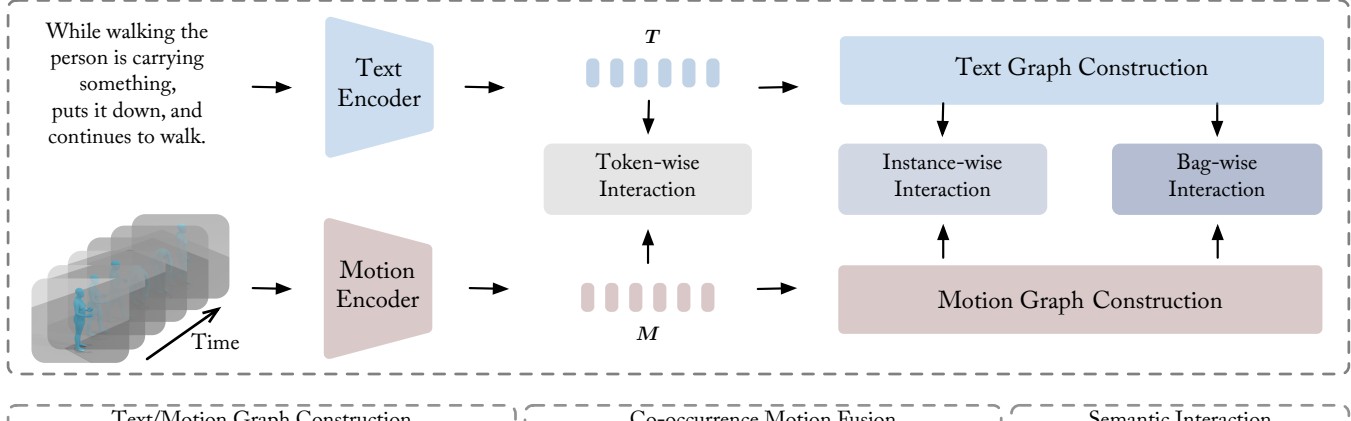

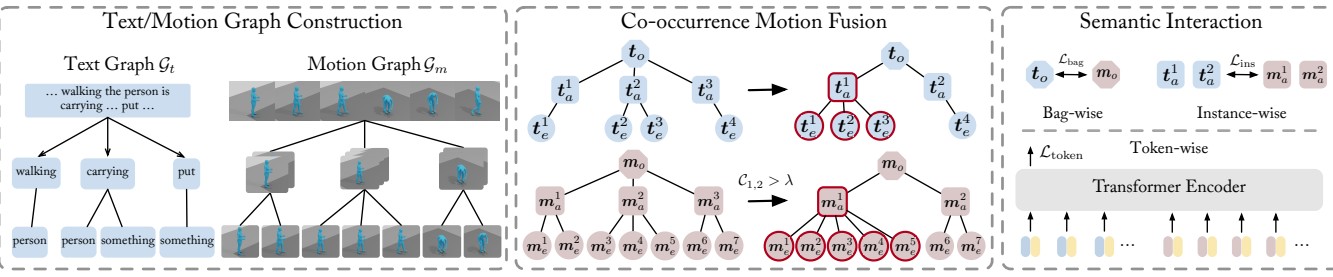

Figure 2: The overview of our multi-granularity semantics interaction (MGSI). Initially, we adopt the text and motion encoder to encode text and motion sequence into word and frame-level embeddings $T$ and $M$, respectively. Then, we decompose the text and motion sequences into different levels by exploiting the graph neural networks to model the semantics correspondence. By adopting the proposed co-occurrence motion mining, the atomic motion $m_a^1$ and $m_a^2$ are fused to formulate the new nodes. Finally, we exploit token, instance, and bag-wise semantics interaction to comprehensively align text and motions sequence.

into a common space and exploits the cross-modal distance, *e.g.*, cosine similarity $\cos(t, m)$, to measure and rank the semantics similarity achieving text-motion retrieval. However, we argue that the motion sequence consists of multiple atomic motions with diverse semantics, the simplistic point representations hardly capture the complicated semantics of motion sequence.

In this work, we formulate the text-motion retrieval as the multi-instance multi-label learning (MIML) problem, where the motion sequence and query are viewed as the bags of multiple instances. For motion and text bags, the instances are viewed as the atomic motions and corresponding descriptions. To solve the MIML problem, we propose a novel multi-granularity semantics interaction (MGSI) approach that exploits the graph neural networks (GNNs) to model the text and motion sequence into different levels, performing the corresponding level semantics interaction to achieve a precise alignment and considerable performance. Specifically, we decompose the text and motion sequences into different level and represent them as the graph $\mathcal{G}_t = \{V_t, E_t\}$ and $\mathcal{G}_m = \{V_m, E_m\}$, where $V_t = \{t_o, \{t_a^i\}_{i=1}^{N_a}, \{t_e^i\}_{i=1}^{N_e^t}\}$ and $V_m = \{m_o, \{m_a^i\}_{i=1}^{N_a}, \{m_e^i\}_{i=1}^{N_e^m}\}$, $N_*$ indicates the number of each type nodes. The $t_o$ and $m_o$ are the root nodes containing the sentence and sequence-level semantics. On the textual side, the instance nodes $\{t_a^i\}_{i=1}^{N_a}$ are the verbs, and the token nodes $\{t_e^i\}_{i=1}^{N_e^t}$ are the noun phrases. For the motion side, the instance nodes $\{m_a^i\}_{i=1}^{N_a}$ indicates the representations of atomic motions, and the token nodes $\{m_e^i\}_{i=1}^{N_e^m}$ are each frame of atomic

motion. The edge $E_t$ and $E_m$ are the edge embeddings that are illustrated in the Sec. 4. After that, we perform graph reasoning to aggregate the semantics and semantics interaction in different level to align the text and motion sequence achieving the comprehensive cross-modal alignment. The overview of our methods is shown in Fig. 2.

## 4 METHODOLOGY

### 4.1 Text Graph Construction

Given a natural language description $T = \{t_1, \cdots, t_{N_t}\}$ with $N_t$ words, we adopt a pre-trained frozen CLIP text encoder [18] to encode $T$ as a word sequences $T = \{t_1, \cdots, t_{N_t}\}$, where $t_* \in \mathbb{R}^{d_t}$. We perform mean pooling for $T$ to initialize the $t_o$. Then, we adopt the off-the-shelf semantic role parser [20] to extract noun phrases and verbs from $T$ as well as their semantic role $E_t$ of each noun phrase. The verbs representations are leveraged as the instance nodes $\{t_a^i\}_{i=1}^{N_a}$ and connected with the root node $t_o$ with direct edges. The noun phrases are used as the token nodes $\{t_e^i\}_{i=1}^{N_e^t}$ and connected with corresponding instance nodes $t_a^*$, where the edge $e_{i,j}$ between $i$-th token node and $j$-th instance node is represented by the semantics role of the token about the motion. Considering that multiple atomic motions may occur simultaneously to the same token node, we duplicate the token node for each semantic role and connect them with corresponding motion nodes. In Fig. 2, we also show the example of the constructed text graph.

## 4.2 Motion Graph Construction

For the motion sequence $M$, we adopt the the model in previous work [4] to extract the skeleton features $M_s \in \mathbb{R}^{N_m \times J \times d_s}$, where J and $d_s$ are the number and feature dimension of skeleton nodes, respectively. Then, the SMPL [10] is adopted to extract the frame-level representations $M_f = \{m_1, \cdots, m_{N_f}\}$ from $M_s$, where the $N_f$ indicates the frame number. Before constructing the instance node, we first downsample the $M_f$ in the temporal domain to reduce the length of the feature sequence. It reduces the computational complexity while maintaining comparable performance. For motion sequence representation $M_f \in \mathbb{R}^{N_f \times d_m}$, we downsample it into a fixed number of features $M_d \in \mathbb{R}^{N_m \times d_m}$ by conducting the mean pooling to prevent lose information of these reduced frames, where $N_m < N_f$ is the number of frames after downsampling.

**Graph Initialization.** We apply mean pooling to aggregate semantics from $M_f$ and get the sequence-level representations to initialize $m_o$. For the instance nodes, considering the motions occur sequentially, we adopt the simple yet effective slide windows strategy [3] to construct the atomic motion instance. Specifically, we set up multiple slide windows of different lengths with a stride of 1 and perform overlapping as the sliding windows move. The windows size set to $w = \{1, 2, \cdots, N_m\}$. Given a sliding window $k$, a clip feature is obtained by mean pooling over the features within $k$. The motion sequences could be split as the motion clips $M_c = \{m_c^i\}_{i=1}^{N_c}$, where $M_c \in \mathbb{R}^{N_c \times d_m}$ and $N_c = \frac{N_m(N_m+1)}{2}$. To remove the redundant clips from $M_c$, we exploit the previously constructed probabilistic embedding space [14] to filter the clips:

$$m_a^i = \max \left\{ \cos(\hat{t}_a^i, \hat{m}_c^1), \cos(\hat{t}_a^i, \hat{m}_c^2), \cdots, (\hat{t}_a^i, \hat{m}_c^{N_c}) \right\} \quad (1)$$

where $\hat{t}_a^i$ and $\hat{m}_c^j$ are the $i$-th noun phrase and $j$-th clips that both are extracted by the pre-trained model [14]. Then, we adopt $\{m_a^1, \cdots, m_a^k\}$ to initialize the embeddings of instance nodes. For token node $m_e^i$, we directly adopt the frame-level representations of each atomic motion as the embeddings to initialize. The instance nodes connect to the root and token nodes with edges that is calculated by:

$$e_{r,i} = \cos(m_a^i, m_*^r) \quad (2)$$

where $\cos(\cdot, \cdot)$ indicates the cosine similarity. The $* \in \{e, o\}$ and $r$ indicates the node index associated with node $i$.

## 4.3 Co-occurrence Motion Fusion.

In TMR, the atomic motion in the sequence may be co-occurring and coupled together. Directly representing these semantically complex data samples as point embeddings and performing alignment may be unable to capture the abundant semantics and disturb the cross-modal alignment affecting the retrieval performance. To address this challenge, we introduce a co-occurring motion mining approach by measuring the semantic consistency between atomic motions. Then, we fuse the identified co-occurring motions to generate a new co-occurrence motion representation. Specifically, given the motion graph $G^m = \{V_m, E_m\}$, the token nodes $\{m_e^x\}_{x=1}^{N_e^t}$ connected to the instance nodes $m_a^i$ are used for calculating the semantics

consistency score $C_{i,j}$:

$$C_{i,j} = \frac{1}{|m_a^i||m_a^j|} \sum_x^{|m_a^i|} \sum_y^{|m_a^j|} \cos(m_e^x, m_e^y) \quad (3)$$

where $|m_a^i|$ and $|m_a^j|$ indicate the degree of $m_a^i$ and $m_a^j$, respectively. The Eq. 3 draws inspiration from the principle that co-occurring motions may encapsulate as many semantically similar frames as possible. After calculating the semantics consistency score, we empirically set a threshold $\lambda$ to mine the co-occurring motions. If the semantics consistency score $C_{woi,j}$ is larger than the threshold $\lambda$, the atomic motions $i$ and $j$ are identified as the co-occurring motions, otherwise are considered as the motions that occurred sequentially. For these co-occurring motions, the nodes $m_a^i$ and $m_a^j$ are fused to obtain the co-occurrence motions nodes by adding and all token nodes belonging to $m_a^i$ and $m_a^j$ are connected to the new instance nodes. As shown in Fig. 2 (b), the $C_{1,2}$ is larger than the threshold $\lambda$. Therefore, the instance nodes $m_a^1$ and $m_a^1$ are merged as the newly instance nodes $m_a^1$, where all corresponding token nodes are connected with the new nodes. Similarly, the corresponding instance nodes in the textual graph $G_t$ are fused to guarantee the structure consistent with $G_m$. Notable, we utilize the residual connection in graph reasoning to reduce the redundancy information introduced by these token nodes.

## 4.4 Graph Reasoning

**Text Graph Reasoning.** Considering the multiple semantics role involved in text, we adopt the rational graph convolutional network (R-GCN) [19] to model correlations of these nodes. Specifically, considering the existence of two types of nodes in the text graph, we adopt 2-layer graph convolution networks with residual connection to capture the semantics of nodes. Based on the initialized nodes $V_i^t = \{t_o, \{t_a^i\}_{i=1}^{N_a}, \{t_e^i\}_{i=1}^{N_e^t}\}$ and the correlation edge $E_t = \{e_{r,i}\}$, the node feature is aggregated by:

$$\begin{aligned} H_i^{t,1} &= \text{ReLU}(\sum_{r \in R} e_{r,i} V_i^t \cdot W_r^{t,1} + V_i^t) \\ H_i^{t,2} &= \text{ReLU}(\sum_{r \in R} e_{r,i} H_i^{t,1} \cdot W_r^{t,2} + H_i^{t,1}) \end{aligned} \quad (4)$$

where $e_{r,i}$ indicates the semantic role of node $i$. The $R$ indicates the number of relations of node $i$. The ReLU($\cdot$) is the ReLU activation function [1]. The $W_r^{t,*}$ are the learnable parameters.

**Motion Graph Reasoning.** Therefore, we obtain the nodes in the motion graph $V_i^m = \{m_o, \{m_a^i\}_{i=1}^{N_a}, \{m_e^i\}_{i=1}^{N_e^m}\}$ Similar to the textual graph reasoning, we adopt 2-layer graph neural networks with residual to aggregate the semantics from the neighbor nodes:

$$\begin{aligned} H_i^{m,1} &= \text{ReLU}(\sum_{r \in R} e_{r,i} V_i^m \cdot W_r^{m,1} + V_i^m) \\ H_i^{m,2} &= \text{ReLU}(\sum_{r \in R} e_{r,i} H_i^{m,1} \cdot W_r^{m,2} + H_i^{m,1}) \end{aligned} \quad (5)$$

where the $W_r^{m,*}$ are the learnable parameters of the motion graph. The $R$ is number of relations corresponding to node $i$.

## 4.5 Semantics Interactions

**Token-wise Interaction.** Besides the instance and bag-wise semantics interaction, we introduce a toke-wise interaction to provide fine-grained semantics alignment as complementary. As shown in Fig. 2(b), we add the position embeddings $p_t$ and $p_m$ to the word- and frame-level representaions $T$ and $M$ and feed them into a transformer encoder:

$$X_{\text{token}} = \Phi_t([T + p_t; M + p_m]) \tag{6}$$

where $[\cdot; \cdot]$ is the concatenate operation. $\Phi_t$ is the 2-layer transformer encoder. We further adopt multi layer perception (MLP) with ReLU activation function [1] to calculate the similarity:

$$S_{\text{token}} = \text{softmax}\left(\text{MLP}\left(X_{\text{token}}[0, :]\right)\right) \tag{7}$$

where the MLP consists of two linear layers with ReLU activation functions. **Instance-wise Interaction.** We adopt the representation of instance nodes from the text and motion graph to perform the instance-wise semantics interaction. Specifically, given the $\{t_a^i\}_{i=1}^{N_a}$ and $\{m_a^i\}_{i=1}^{N_a}$, the instance-wise similarity is calculated by:

$$S_{\text{ins}} = \frac{1}{N_a} \sum_{i=1}^{N_a} \cos(t_a^i, m_a^i) \tag{8}$$

**Bag-wise Interaction.** For the bag-wise semantics interaction, we directly conduct the semantic interaction between the root node representations of text and motion graph:

$$S_{\text{bag}} = \cos(t_o, m_o) \tag{9}$$

## 4.6 Model Training and Inference

**Training.** In our proposed multi-granularity multi-instance learning, the positive pair is defined as the motion containing certain content relevant to the query text. The negative pairs are those without any relevant content. We adopt the InfoNCE loss [28, 29] function that is widely used in the retrieve related tasks as the training objective function over the mini-batch $B$:

$$\mathcal{L}_{\text{info}} = -\frac{1}{|B|} \sum_{x_i, y_i \in B} \left[ \log \frac{S(x_i, y_i)}{S(x_i, y_i) + \sum_{i \neq j} S(x_i, y_j)} + \frac{S(y_i, x_i)}{S(y_i, x_i) + \sum_{i \neq j} S(y_i, x_j)} \right] \tag{10}$$

where $S(\cdot, \cdot)$ is the similarity measurement, *e.g.* cosine similarity. The token-wise semantics interaction is defined as follows:

$$\mathcal{L}_{\text{token}} = \text{CrossEn}(S_{\text{token}}) \tag{11}$$

where CrossEn is the cross entropy loss function. Thus, the total training loss function is:

$$\mathcal{L}_{\text{train}} = \mathcal{L}_{\text{token}} + \mathcal{L}_{\text{ins}} + \mathcal{L}_{\text{bag}} \tag{12}$$

where $\mathcal{L}_{\text{ins}}$ and $\mathcal{L}_{\text{bag}}$ indicate applying the instance-wise similarity $S_{\text{ins}}$ and bag-wise similarity $S_{\text{bag}}$ to the Eq 10, respectively.
**Inference**. After the model is converged, the similarity between query text $T$ and motion sequence $M$ is computed by a combination of the instance-wise, bag-wise, and token-wise similarities:

$$S(T, M) = \frac{1}{3}\left(S_{\text{token}} + S_{\text{ins}} + S_{\text{bag}}\right) \tag{13}$$

## 5 EXPERIMENTS

### 5.1 Datasets

We validate the proposed methods on the two widely used 3D human motion datasets: HumanML3D [4] and KIT Motion-Language Datasets [16]:
**HumanML3D** [4] (HumanML3D) is currently the largest 3D human motion dataset with textual descriptions. The motion sequences are originally from two already-existing and widely-used motion-capture datasets AMASS [11] and HumanAct12 [5]. Following the benchmark [15], we split the train, validation, and test set with 23384, 1460, and 4380 motions. Each motion sequence contains approximately 3 text descriptions with different lengths.
**KIT Motion-Language** [16] (KIT-ML) contains 3,911 recordings of full body motion and 6,278 text descriptions. Each motion sequence is described in 1 to 4 texts. The average length of text descriptions is approximately 8. Following the setup in the benchmark [15], we adopt 4,888, 300, and 800 motion sequences as the training, validation, and test set, respectively.

### 5.2 Baselines and Metrics

**Baselines**. We provide the comprehensive comparison with the state-of-the-art approaches, including TEMOS (ECCV2022) [14], MotionCLIP (ECCV2022) [21], T2M (CVPR2022) [4], DTL (MM Asia2023) [24], TMR (ICCV2023) [15], and MoT (SIGIR2023) [12]. The MotionCLIP [21], T2M [4], and MoT [12] learn the text and motion into a common space, directly measuring the cosine similarity between global embeddings for alignment. The TEMOS [14] and TMR [15] employ a VAE structure to learn the text and motion into latent space while adopting the reparameterization technique to sample the representations from distributions for alignment [14, 15]. The DTL [24] adopts a dual-branch unimodal network to extract motion and text embeddings and project them into a common embeddings space. However, the DTL splits the HumanML3D and KIT-ML by themselves and the scale of test sets is far less than the splits used in our work. To compare the results fairly, we adopt the open-source code of DTL to train the model in our splits and report the results. The other results of baselines in our work are from their official reports.
**Metrics**. We adopt the common metrics to report retrieval performance, including Recall at K (R@K), Median Rank (MedR), and Rsum. The R@K is the fraction of queries that correctly retrieve desired items in the top K of the ranking list. Following the benchmark [15], K = 1,2,3,5,10 are adopted. The MedR computes the median rank of the correct targets for a query. Additionally, we report the Rsum metric which is calculated by the summing of R@K values. It evaluates retrieval performance from an overall perspective. In all tables, the metric with an upward arrow (denoted by ↑) signifies that a higher value correlates with better performance (R@K, Rsum), while the downward arrow (denoted by ↓) indicates that lower values represent superior performance (MedR). The best evaluation results are highlighted in "**bold**".

### 5.3 Implementation Details

In this work, we adopt AdamW [8] as the optimizer with a 1e-4 learning rate and set the batch size to 64 on all datasets. The text

**Table 1: Performance comparison with the state-of-the-art methods on HumanML3D [4]. The "Text → Motion" indicates text-to-motion retrieval and "Motion → Text" is the motion-to-text retrieval, respectively.**

| Methods | Text → Motion | | | | | | Motion → Text | | | | | | Rsum ↑ |
|---|---|---|---|---|---|---|---|---|---|---|---|---|---|
| | R@1 ↑ | R@2 ↑ | R@3 ↑ | R@5 ↑ | R@10 ↑ | MedR ↓ | R@1 ↑ | R@2 ↑ | R@3 ↑ | R@5 ↑ | R@10 ↑ | MedR ↓ | |
| T2M [4] | 1.80 | 3.42 | 4.79 | 7.12 | 12.47 | 81.00 | 2.92 | 3.74 | 6.00 | 8.36 | 12.95 | 81.50 | 63.57 |
| TEMOS [14] | 2.12 | 4.09 | 5.87 | 8.26 | 13.52 | 173.00 | 3.86 | 4.54 | 6.94 | 9.38 | 14.00 | 183.25 | 72.58 |
| MotionCLIP [21] | 2.33 | 5.85 | 8.93 | 12.77 | 18.14 | 103.00 | 5.12 | 6.97 | 8.35 | 12.46 | 19.02 | 91.42 | 99.94 |
| MoT [12] | 2.61 | 4.72 | 6.90 | 10.66 | 17.79 | 60.00 | 4.03 | 5.07 | 7.43 | 11.23 | 17.68 | 64.25 | 88.12 |
| DTL [24] | 2.69 | 4.93 | 7.42 | 11.36 | 17.71 | 73.00 | 2.33 | 4.50 | 6.50 | 10.31 | 17.48 | 76.00 | 85.24 |
| TMR [15] | 5.68 | 10.59 | 14.04 | 20.34 | 30.94 | 28.00 | 9.95 | 12.44 | 17.95 | 23.56 | 32.69 | 28.50 | 178.18 |
| MGSI (Our) | **6.61** | **12.73** | **17.11** | **23.91** | **34.74** | **24.00** | **10.61** | **13.18** | **19.75** | **26.00** | **36.63** | **22.50** | **201.27** |

**Table 2: Performance comparison with the state-of-the-art methods on KIT-ML [16]. The "Text → Motion" indicates text-to-motion retrieval and "Motion → Text" is the motion-to-text retrieval, respectively.**

| Methods | Text → Motion | | | | | | Motion → Text | | | | | | Rsum ↑ |
|---|---|---|---|---|---|---|---|---|---|---|---|---|---|
| | R@1 ↑ | R@2 ↑ | R@3 ↑ | R@5 ↑ | R@10 ↑ | MedR ↓ | R@1 ↑ | R@2 ↑ | R@3 ↑ | R@5 ↑ | R@10 ↑ | MedR ↓ | |
| T2M [4] | 3.37 | 6.99 | 10.84 | 16.87 | 27.71 | 28.00 | 4.94 | 6.51 | 10.72 | 16.14 | 25.30 | 28.50 | 129.39 |
| TEMOS [14] | 7.11 | 13.25 | 17.59 | 24.10 | 35.66 | 24.00 | 11.69 | 15.30 | 20.12 | 26.63 | 36.39 | 26.50 | 207.84 |
| MotionCLIP [21] | 4.87 | 9.31 | 14.36 | 20.09 | 31.57 | 26.00 | 6.55 | 11.28 | 17.12 | 25.48 | 34.97 | 23.00 | 175.60 |
| MoT [12] | 6.23 | 11.07 | 16.54 | 23.92 | 37.15 | 20.00 | 10.56 | 13.49 | 20.61 | 27.61 | 38.04 | 19.50 | 205.22 |
| DTL [24] | 8.07 | 13.28 | 16.92 | 22.91 | 36.97 | 18.00 | 9.11 | 14.84 | 19.27 | 26.04 | 39.06 | 17.00 | 206.51 |
| TMR [15] | 7.23 | 13.98 | 20.36 | 28.31 | 40.12 | 17.00 | 11.20 | 13.86 | 20.12 | 28.07 | 38.55 | 18.00 | 221.80 |
| MGSI (Our) | **8.91** | **16.28** | **20.87** | **29.64** | **40.84** | **16.00** | **13.49** | **16.41** | **23.54** | **30.66** | **43.00** | **15.50** | **243.64** |

representations dimension $d_t$, and the motion sequence dimension $d_m$ are set to 256 and 263, respectively. The downsampled frame number $N_m$ is 200. The GNNs learned cross-modal common space dimension is set to 256. The hyperparameter $\lambda$ for co-occurrence motion filtering is empirically set to 0.8. Our experiments are implemented in PyTorch-1.10 and are conducted on 8 NVIDIA A800 GPUs with 80GB memory. To enable a consistent comparison with the baseline, we follow the settings of previous work [15] to randomly select a text as the matching text for training and adopt the first text in the test set to report the evaluation performance.

## 5.4 Performance Comparisons

In this subsection, we show the experimental results of the proposed MGSI and the state-of-the-art (SOTA) methods of TMR on the HumanML3D [4] and KIT-ML [16]. In Tab. 1, we observe that the baselines [4, 12, 14, 21] show an unsatisfying performance. The conventional approaches roughly represent the query and motion sequences as a single point to perform a global-level alignment, which are based on the assumption that the data instance in TMR only involves unique semantics. However, in reality, there exist many different motions in the query and motions sequences. Such simplistic learning strategies may difficult to capture the complex semantics resulting in an unsatisfying performance especially on the fine-grained retrieval (R@1). In our work, we formulate the

TMR as the multi-instance multi-label learning trying to decompose the coupled semantics in the query and motion sequences and perform semantics alignment in corresponding levels. The significant improvement (23.09% in Rsum) of our MGSI in HumanML3D proves the effectiveness of our methods.

For the KIT-ML dataset, it shows that our MGSI surpasses the current SOTA TMR approaches across all evaluation protocols with a clear margin. Especially on the Rsum metrics, our method outperforms the SOTA work TMR [15] by 21.84%. Since these baselines focus on the whole similarity between queries and motion sequences, the result indicates that such coarse-grained global similarity modeling is sub-optimal for TMR. It demonstrates the superiority of our hierarchical semantics alignment in text to 3D human motion retrieval.

## 5.5 Ablation Study

**The effectiveness of semantics interactions**. In this subsection, we verify the contributions of the different levels semantics interactions in the proposed multi-instance multi-label learning. As shown in Tab. 3, we observe that 1) when applying the bag-wise semantics interaction, the model shows the worst performance, which is close to the baselines solely aligning the point embeddings of text and motion (in Tab. 1 and Tab. 2). It proves that the simplistic global semantics alignment is unsuitable for the TMR. 2) When incorporating the instance- or token-wise with bag-wise

**Table 3: Ablation study of different semantics interaction on HumanML3D and KIT-ML. We report the results on the text-to-motion retrieval.**

| $\mathcal{L}_{token}$ | $\mathcal{L}_{ins}$ | $\mathcal{L}_{bag}$ | HumanML3D | | | KIT-ML | | |
|---|---|---|---|---|---|---|---|---|
| | | | R@1 ↑ | R@2 ↑ | R@3 ↑ | R@1 ↑ | R@2 ↑ | R@3 ↑ |
| ✗ | ✗ | ✓ | 2.53 | 4.71 | 8.30 | 5.18 | 10.77 | 15.92 |
| ✗ | ✓ | ✓ | 5.89 | 11.34 | 15.31 | 8.02 | 16.41 | 20.74 |
| ✓ | ✗ | ✓ | 5.63 | 10.83 | 14.94 | 7.25 | 14.89 | 20.10 |
| ✓ | ✓ | ✓ | 6.61 | 12.73 | 17.11 | 8.91 | 16.28 | 20.87 |

**Table 4: The ablation studies to investigate the proposed each components on the HumanML3D.**

| Methods | Text → Motion | | | Motion → Text | | |
|---|---|---|---|---|---|---|
| | R@1 ↑ | R@2 ↑ | R@3 ↑ | R@1 ↑ | R@2 ↑ | R@3 ↑ |
| w/o downsample | 6.28 | 12.39 | 17.66 | 10.81 | 13.71 | 20.03 |
| w/o motion fusion | 5.97 | 11.80 | 15.79 | 9.41 | 12.11 | 18.35 |
| MGSI | **6.61** | **12.73** | **17.11** | **10.61** | **13.18** | **19.75** |

semantics interaction, the model achieves a considerable retrieval performance. The improvements achieved by the instance- and token-wise semantics interaction indicate the necessity of aligning the query text and motion sequence in a fine-grained scale. 3) The improvements brought by the instance-wise are larger than the token-wise. Considering the redundant information in token, directly concatenating all tokens of text and motion without filtering may introduce too much useless information to disturb the cross-modal aligning. The proposed instance-wise semantics interaction exploits the specifically designed downsample and co-occurring motion mining to refine the semantics within motion sequences bringing much retrieval performance. 4) The complete version of our method, incorporating bag-, instance-, and token-wise semantics interaction, shows the best retrieval performance. This proves that our approach that focuses on semantics interactions at different level is complementary to each other.

**The effectiveness of components**. To examine the usefulness of the specific designed strategies in in MGIS, we compare the counterpart without the downsample or the motion fusion on the HumanML3D. As shown in Tab. 4, we observe that 1) when removing the downsample strategy, the retrieval results are further boosted but limited. Adopting all frame of motion sequences may introduce significant computing costs. Therefore, we conduct and report all experiment results on the downsampling version. 2) The retrieval performance degenerates when we detach the co-occurrence motion fusion and straightly align the initialized instance nodes in the text and motion graph, which proves the effectiveness of our motion fusion strategy.

**The effectiveness of clip selection**. In this subsection, we aim to verify the clip selection strategy in Sec. 4.2. The results are shown in Tab. 5. The "Random" indicates that randomly selecting the clips from $M_c$ constructs the instance nodes in the motion graph. The results show that the clips $M_c$ selected by the sliding window contain useless semantics for retrieval. The simplistic random selection criterion in mining the atomic motion is ineffective. The

**Table 5: The investigation of motion clip selection strategy on HumanML3D.**

| Strategies | Text → Motion | | | Motion → Text | | |
|---|---|---|---|---|---|---|
| | R@1 ↑ | R@2 ↑ | R@3 ↑ | R@1 ↑ | R@2 ↑ | R@3 ↑ |
| Random | 2.11 | 3.59 | 5.13 | 2.01 | 3.22 | 5.48 |
| MGSI | 6.61 | 12.73 | 17.11 | 10.61 | 13.18 | 19.75 |

proposed clips filter strategy in Eq. 1 utilizes the pre-constructed cross-modal probabilistic space to find semantically similar clips to the action phrases. The performance improvement compared to "Random" proves that the proposed methods could effectively filter the irrelevant clips.

**Hyperparameter analysis**. In this subsection, we investigate the influence of hyperparameters $\lambda$ on the HumanML3D. The $\lambda$ is the threshold to identify the co-occurrence motions. If the semantic consistency score is greater than $\lambda$, these actions are considered co-occurring motions. As shown in Fig. 4, we set the $\lambda$ from 0.1 to 0.9. The results lead to several observations: 1) when $\lambda$ is small, the retrieval performance remains lower and improves with the increase of $\lambda$, which proves that a loose criterion is insufficient to identify the accurate co-occurrence motions. The co-occurrence motions should maintain as many semantically consistent frames as possible. 2) When $\lambda$ exceeds the threshold of 0.8, there is a noticeable decline in retrieval performance. The higher value set for $\lambda$ may overly stress the semantic consistency, imposing an overly strict criterion. It could limit the detection of co-occurring motions, potentially reducing the retrieval performance.

## 5.6 Visualization Results

In Fig. 3, we visualize the retrieval results for text-to-motion retrieval of the proposed MGSI and the state-of-the-art TMR [15] on the HumanML3D. For each method, we draw the top-5 retrieved motion sequences, where we rank the results by similarities and give the annotation at the bottom of each motion sequence. We also highlight the semantically similar verbs to the query by the same color. The successful and failed retrieval results are highlighted by the green and red border, respectively. In Fig. 3, the query contains multiple atomic motions with different semantics, especially the motion "walks" and "holds" are co-occurring. For the TMR, we notice that 1) the successful retrieval result is in the fourth position, which means that the TMR can retrieve semantically similar motions, but with limited precision. 2) The first and second returned results only contain only part of the semantics consistent with the query. It may caused by the single-point representation only capturing the simplistic motion semantics ("put" and "walk") and failing to represent the complicated motion sequences. Further aligning these rough representations can significantly disturb the cross-modal alignment and undermine the retrieval performance.

For the results of MGSI, we observe that 1) the proposed method successfully retrieves the motion sequences corresponding to the query by accurately identifying these co-occurring motions and learning semantic correlations between the query and candidate from multiple granularities, demonstrating our methods' effectiveness. 2) The second retrieved motion sequence still contains similar

A person walks forward and holds a large object in their hands, puts the object down and continues to walk.

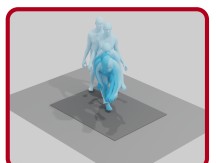 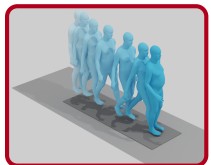 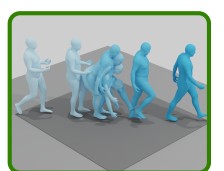 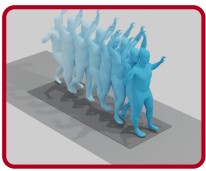

A person put down something and slowly walked forward.

The person walk forward and wipe some thing off.

A person carefully walks down a slope.

A person walks forward and holds a large object in their hands, puts the object down and continues to walk.

A person raises their arms, while walking forward.

TMR

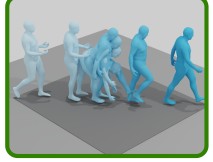 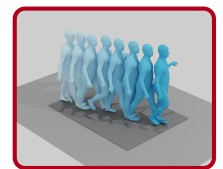 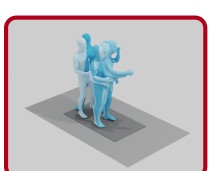 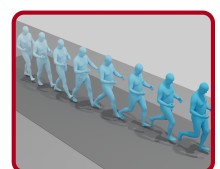 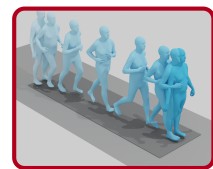

A person walks forward and holds a large object in their hands, puts the object down and continues to walk.

A person holds both hands out and walks slowly forward, putting their hands down by their sides.

A man walks forwards whilst putting his arms in front of him holding something.

A person's walking forward quickly while holding something that they seem to be balancing or positioning.

A person jogs forward and then places their hands at their side.

MGSI

**Figure 3: The visualization of retrieval results. We showcase the top-5 retrieved motion sequences by the proposed MGSI and state-of-the-art TMR, respectively. The sentences below at the motion sequences are the corresponding annotations. We adopt the same color to the query text to highlight the verbs in the results' annotations to help evaluate the retrieval performance. The successful and failed retrieval results are highlighted by the green and red border, respectively.**

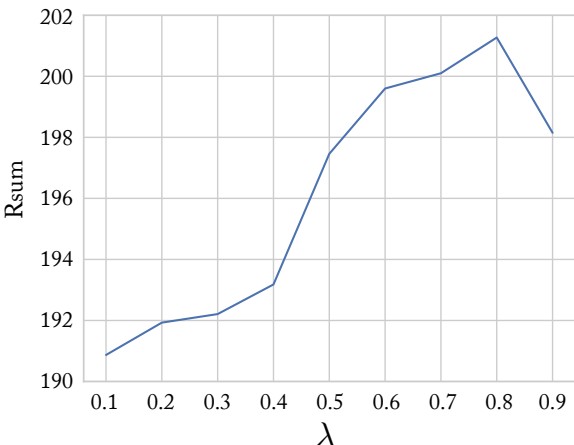

**Figure 4: The experimental result is the investigation of the threshold $\lambda$ in selecting the co-occurrence atomic motions.**

semantics to the query ("hold", "walks", and "putting down"). It demonstrates that our MGSI can successfully capture fine-grained semantics between the text and motion sequences. 3) Although other results are the failed retrieve results, they still contain these co-occurring motions ("walk" and "hold"), which verifies the effectiveness of the proposed co-occurrence motion mining approach. These observations suggest that through integrating token, instance,

and bag-wise semantics interactions, our MGSI can capture both fine-grained and overall semantics, ensuring a comprehensive semantic analysis.

## 6 CONCLUSIONS AND FUTURE WORKS

In this paper, we conceptualize the Text-Motion Retrieval (TMR) task as a Multi-Instance Multi-Label (MIML) learning problem, where each motion sequence is viewed as a bag of atomic motions, and the corresponding text as a bag of phrases. To tackle the MIML challenge within TMR, we introduce a novel Multi-Granularity Semantics Interaction (MGSI) approach, which effectively captures and aligns the semantics of text and motion sequences across various levels. Specifically, the MGSI approach decomposes both query and motion sequences into three hierarchical levels: token, instance, and bag. We then utilize graph neural networks to explicitly model and interact with their semantic correlations at these levels, thus capturing the semantics across multiple granularities accurately. To identify and model co-occurring atomic motions, we measure semantic consistency frame-wise, then fuse and interact the accordant motions to refine their representations. Finally, we employ token, instance, and bag-wise semantic interactions to comprehensively align the text and motion sequences. Extensive experiments on two widely-used datasets demonstrate the efficacy of our methods

In our future work, we plan to incorporate informative skeleton features to enhance precise atomic motion mining, further facilitating fine-grained semantic interaction between text and motion.

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
