# OpenReview forum: "Multi-Instance Multi-Label Learning for Text-motion Retrieval"
_acmmm.org/ACMMM/2024/Conference — MM2024 Poster_

### Official Review · Reviewer_RGpi · 2024-05-24

**Rating:** 4
**Confidence:** 3

**Summary:**

The authors tackle text-motion retrieval using the framework of multi-instance multi-label (MIML) learning. Specifically, they propose to decompose both texts and motions into their building blocks at three different granularity levels (bag, instance, token). Successively, they perform some filtering of the constructed graphs to remove redundant information and fuse instance nodes if they contain too similar token nodes. They then aggregate information at different granularity levels using graph neural networks and enforce three different losses for the information extracted at each level. They perform experiments on two widely employed datasets for text-motion retrieval.

**Strengths:**

- The paper tackles the important problem of separating fine-grained semantics of motions that, differently from images or videos, contain much higher inter-class confusion. The idea of employing a multi-instance learning framework, in this context, is sound.
- The results on both KIT and HumanML3D datasets show the effectiveness of the proposed approach with respect to the baselines, where a consistent improvement margin is obtained.
- The final ablation study validates the components of the proposed architecture.

**Limitations:**

- One of the claims of this paper is that global features approaches used by the baselines are not sufficient to solve this task. This is further remarked in the ablation study in Table 3, where only the bag (global) feature matching obtains a performance comparable with the early baselines. However, even recent approaches (TMR) employ global features but obtain much better results. In this sense, it is strange that the performance of solely the bag components achieves such low results. I would have expected the bag features to be comparable with TMR. This could signify that the global features from the proposed architecture carry too little information compared with other global-based descriptors.
- It is strange to frame the problem using a multi-label framework. While it is clear that it could be multi-instance, it is strange to treat it as multi-label, given that labels are not explicit and only appear in the final natural language sentences. In this sense, it may not be a multi-label task from the standard perspective, which should be better stated in the text.
- The clip selection procedure, as shown in Table 5, seems very important, as the performance triples with respect to the random selection. However, the selection is performed using a pre-trained text-motion model. This means that the proposed method may be highly dependent on one of the baselines (TEMOS in this case). I would expect the authors to better comment on this (e.g., did they try other more performant methods (e.g., TMR [15]) or other selection criteria?)
- Why did the authors use a transformer for the token-wise interaction while instead a simple aggregation of the cosine similarities for bag and instance interactions? Did they also try a simpler aggregation for token-wise interaction?
- There are wrong bolds in Table 4 (the bold ones are not the best values). This means that, in some cases, the best results are not the ones where all the modules collaborate to reach the best performance. In the light of this, a more careful discussion in the ablation studies is needed.

Minor issues:
- I believe S_{token} in Eq. 7 is obtained through a sigmoid and not a softmax from the output of the MLP (I assume the MLP output is 1-dimensional). If so, I also assume cross-entropy in Eq. 11 is a binary cross-entropy loss function. If so, please change softmax to sigmoid in Eqs. 7 and specify that is a binary CE in Eq. 11.
- The paragraph title “Instance-wise Interaction” in Section 4.5 should be on a new line.

**Suitability:**

3

---

### Official Review · Reviewer_ZBbd · 2024-05-27

**Rating:** 6
**Confidence:** 2

**Summary:**

This paper focus on the challenge that co-occurring and coupling of atomic actions with complex semantics is hard to be represented by single
embeddings in the text-motion retrieval task. It formulates TMR as a Multi-Instance Multi-Label learning problem and  propose a novel Multi-Granularity Semantics Interaction (MGSI) approach, which effectively captures and aligns the semantics of text and motion sequences across various levels.

 Overall,  this is a valuable work in my opinions. It offers a new perspective to consider the text-motion retrieval problem from the viewpoint of MIML. The proposed framework appears to be coherent and logically structured. It aligned semantics from the perspectives of tokens, instances, and bags respectively. The experiment also appears to be sufficient and convincing

**Strengths:**

1.  This paper formulates TMR as a Multi-Instance Multi-Label(MIML) learning problem which is the first attempt in this field.
2.  It proposes a novel Multi-Granularity Semantics Interaction (MGSI) approach to explicitly model the semantics in 3 levels and estabilished their semantic consistency.
3. The obvious improvement on experiment results shows their efficacy.

**Limitations:**

1. The result in Figure 4  seems to show the model is sensitive on \lambda. Is the value of this hyperparameter sensitive to the dataset? Do different datasets require independent settings? At least a discussion is needed.

**Suitability:**

3

---

### Official Review · Reviewer_q64E · 2024-06-04

**Rating:** 4
**Confidence:** 2

**Summary:**

This paper aims to address the problem of capturing complex semantics of motion in text-motion retrieval task by treating it as a Multi-instance Multi-label (MIML) learning problem. The authors introduce a Graph Neural Network (GNN)-based Multi-Granularity Semantics Interaction (MGSI) approach which constructs both a text graph and a motion graph. The text graph includes noun phrases, verbs, and bag-wise node, while the motion graph consists of frames, atomic motions, and bag-wise node. Interactions between the corresponding nodes of the two graphs capture multi-level semantic information. Experiments conducted on two commonly used datasets have achieved SOTA performance.

**Strengths:**

1. **Motivation:** The motivation behind this paper is quite clear.
2. **Experiments:** This work has conducted adequate experiments to validate the effectiveness of the proposed method.

**Limitations:**

1. **Novelty:** Many researchs in retrieval-related tasks have shown that more interactions lead to better performance but also result in higher computational costs. The way the authors improves the TMR task may lack novelty. Additionally, the method proposed requires constructing a motion graph based on the text query, which means that motion sequences cannot be pre-processed, leading to potentially high latency. A comparison of latency should be provided. Here are some related references:
     - Khattab, O., & Zaharia, M. ColBERT: Efficient and Effective Passage Search via Contextualized Late Interaction over BERT, SIGIR 2020.
     - Samuel Humeau, Kurt Shuster, Marie-Anne Lachaux, Jason Weston. Poly-encoders：Architectures and Pre training Strategies for Fast and Accurate Multi sentence Scoring, ICLR 2020.

2. **Writing:** There are a few minor points that are unclear.
      **a.** In the explanation of Equation 1, the author mentions the set ${𝑚_𝑎^1,…,𝑚_𝑎^𝑘}$ but does not clarify what 'k' represents. Does the 'k' refer to the number of noun phrases?
      **b.** In Table 4, removing the downsample strategy resulted in a worse performance for text → motion, which contradicts the statement in the paper that "when removing the downsample strategy, the retrieval results are further boosted but limited."

**Suitability:**

3

---

### Official Review · Reviewer_bJP5 · 2024-06-06

**Rating:** 4
**Confidence:** 4

**Summary:**

This paper focuses on enhancing the representation learning for the text-to-motion retrieval task. The assumption is that a motion consists of several atomic sub-motions with complex semantics and, therefore, the motion is viewed as a bag of sub-motions, and the text as a bag of corresponding phrases. I generally like the main paper idea to represent motions as well as text in a graph-based structure and apply co-occurrence motion fusion, instead of learning global text and motion embeddings.

The main concern I have is the experimental evaluation:

1) The TMR [15] paper defines four evaluation protocols (All, All with threshold, Dissimilar subset, Small batches). However, these four protocols are not ever considered in this paper. Thus, it is not clear which protocol was really used for experimenting and whether the accuracy of the proposed approach can be considered better than TMR (it can be better only on some protocols)...

2) The proposed approach is better on HumanML3D and only slightly better on KIT-ML, when comparing individual R@x values in Table 1 and Table 2.

3) The authors mention that: "To compare the results fairly, we adopt the open-source code of DTL to train the model in our splits and report the results.", so they have re-evaluated the results for the DTL approach. However, how have they obtained the results for MoT [12]??? The MoT paper [12] (as the first text-to-motion retrieval approach) does not provide direct results for the motion-to-text scenario, so the authors have probably extracted the implementation of MoT (motion transformer) from the provided source codes and integrated it into the evaluation pipeline, haven't they? Btw, the presented results of MoT are lower on the text-to-motion scenario compared to the results reported in [12]. This is not commented on in the paper.

4) In Table 4, most of the best results (all on the "Motion->Text" scenario) belong to the baseline, i.e., w/o downsample and not to MGSI as wrongly highlighted by bold font.

5) The construction of the graph using a "brute-force" approach (i.e., testing all possible sliding windows) in Section 4.2 should be described and motivated in more detail. Since this approach is computationally expensive, I ask for adding some quantitative results -- at least report the total pre-processing/training times.


Text comments:

At the beginning of Introduction, I miss a brief introduction about what motion data are, what are their applications, and a broader context of the retrieval operation in the context of motion data, as surveyed, for example, in:

Sedmidubsky, J., Elias, P., Budikova, P., Zezula, P.: Content-Based Management of Human Motion Data: Survey and Challenges. IEEE Access, 9, 2021.
https://ieeexplore.ieee.org/document/9416451

At the same time, Introduction can be shortened since the main paper idea "...motion sequence is a bag of atomic motions and the query text is a bag of phrases corresponding to atomic motions..." is repeated several times in the introduction. I do not also like that the content of the first paragraph of Introduction overlaps quite a lot with the content of Section 2.1.

In Section 2.2, the text content starting from the sentence "In this work, ..." is already known and again repeated; however, the principal difference between the proposed MIML approach and existing MIML methods is not explained. Similarly, the content of Section 3 contains a lot of information that is already known. This unnecessarily prolongs the text and distracts from important things. However, I would appreciate a better explanation of many symbols used to define the graph G_t (in Section 3) and also the symbols used in Section 4.2. I also ask for a more detailed description of the training procedure in Section 4.6, which is not very clear. Why do not use terms "t_i"/"m_i" instead of undefined terms "x_i" and "y_i" in Equation 10?


Other comments:
* in the paper title: "... for Text-motion Retrieval" --> "... for Text-Motion Retrieval"
* notation is sometimes confusing, e.g., the definition of the motion clips M_c
* "..., the instance nodes m^1_a and m^1_a..." --> "..., the instance nodes m^1_a and m^2_a..."
* quite a lot of misspellings/errors, e.g., "toke-wise" --> "token-wise", "a effective method" --> "an effective method", "the the model" --> "the model", "multi layer perception" --> "multi layer perceptron", "in the retrieve related tasks" --> "in the retrieval-related tasks", "efficacy of our methods" --> "efficacy of our methods."

**Strengths:**

The main paper idea is interesting and new in the context of text-to-motion retrieval, i.e., representing motions and text as bags of atomic motions/phrases and applying co-occurrence motion fusion, instead of learning global text and motion embeddings.

**Limitations:**

Experimental evaluation is principally well-designed, but there are some misunderstandings. The text description, as well as the formal notation, should be enhanced. See the Summary part for more details.

**Suitability:**

3

---

### Meta-Review · Area_Chair_pmVx · 2024-07-02

**Recommendation:** Accept (Poster)
**Confidence:** 4

**Metareview:**

There is a consensus to accept this paper. The innovation and rich experimentation were appreciated.
The authors provided satisfactory responses to the reviewers' comments, providing reasonable justifications for their design choices.
If the paper is accepted, the authors will have to update the paper as mentioned in the rebuttal.